# The Implication of Benzene–Ethanol Extractive on Mechanical Properties of Waterborne Coating and Wood Cell Wall by Nanoindentation

**Yan Wu [1,2,\*], Yingchun Sun [1,2], Feng Yang [3,\*], Haiqiao Zhang [1,2] and Yajing Wang [1,2]**

1   College of Furnishings and Industrial Design, Nanjing Forestry University, Nanjing 210037, China
2   Co-Innovation Center of Efficient Processing and Utilization of Forest Resources, Nanjing Forestry University, Nanjing 210037, China
3   Fashion Accessory Art and Engineering College, Beijing Institute of Fashion Technology, Beijing 100029, China
\*   Correspondence: wuyan@njfu.edu.cn (Y.W.); yangfeng@bift.edu.cn (F.Y.)

**Abstract:** The waterborne coating uses water as its solvent, which will partially dissolve wood extractives when it is applied to wood surfaces. This influences both the coating curing process and the mechanical properties of the cured coating. To investigate these influences, the mechanical properties of waterborne polyacrylic coating on control and extractive-free wood surfaces were investigated by nanoindentation. Reductions to elastic modulus ($E_r$) and hardness ($H$) of the coating layer was observed in the wood cell walls adjacent to or away from coating layers. Extraction treatment resulted in significant decrease of the $E_r$ and $H$ of the coating layer on extractive-free wood surface comparing with control wood, but the values slightly increased for extractive-free wood cell walls compared to a control. $E_r$ and $H$ of coating in wood cell lumen were higher than the average value of coating layer on wood surface in both the control and extractive-free wood. The $E_r$ of wood cell wall without coating filled in lumen was significantly higher than those of filling with coating. However, there was no distinct difference of $H$. The $E_r$ and $H$ of CCML in extractive-free wood were 15% and 6% lower than those in control ones, respectively.

**Keywords:** benzene–ethanol extractive; waterborne coating; pine wood; wood cell wall; nanoindentation

## 1. Introduction

Wood extractives are the un-bound organic molecules present in wood that dissolve and therefore can be removed using fluxed solvents such as ethanol, benzene, ether, acetone, dichloromethane, or solutions containing set proportions of the aforementioned solvents. These organic species are found in large quantities within resin channels, gum channels, and parenchyma cells in woody biomass. The interactions between the various extractive components and the fixed elements of wood cell walls are extremely complex, in part due to the wide variety of extractives compositions. Examples of literature seeking to better understand the variety of extractives components include Liu et al. [1], who determined the chemical components of benzene/ethanol extractive-free of beech wood using an exhaustive set of GC-MS analyses. Meanwhile, Zhang et al. [2] has studied on benzene/ethanol extractives of eucalyptus urophylla chips using a more complex Py-GC-MS configuration.

The effective extraction and utilization of wood extractives prior to processing not only saves the cost input of wood processing, pulping, and paper making, but also provides cheap and abundant raw materials for medicine, light industry, and chemical industry [3]. Pine wood must be subjected to extractives removal ("degreasing") before processing, because these chemicals negatively affect the

coating efficiency when applying adhesives properties for the variety of solid products manufactured from pine wood. In order to improve the decorative function of Pinus massoniana wood, a neutral degreasing has been carried out with hot ethanol solution [4]. However, the effect of extractives on the mechanical properties of an applied coating and how it interacts with wood cell walls was not established.

As wood and wood-based composites become increasingly important to society's demanding sustainable materials, it has become of great importance to understand such material's overall properties as a function of its constituent's individual interactions at interfacial regions within the solid matrix. One analytical technique used for probing such interactions is nanoindentation, a well-recognized technique that facilitates measurement of micro- or nano-scale mechanical properties of solid materials [5]. The principle is that a particular shape of the pressure head with a vertical pressure into the sample, according to the unloading after the indentation photo to obtain the material surface indentation. Recent developments in such techniques have generated new usefulness for indentation tests at the cell-wall level of bio-based materials [6]. Nanoindentation can be used to better understand these defects and to explore the mechanical interactions between extracts, wood and applied coatings.

Over the past decade, a large number of biomaterials have had insightful nanoindentation research performed up on them, including wood [7–9], bamboo [10], crop stalks [11], and wood-adhesive bond lines [12]. However, detailed information is lacking concerning the physical nature of how an applied coating interacts with both the wood cell lumina and walls which it is directly exposed. Waterborne coatings like UV-cured, acrylic varnish, etc. are increasingly used in wood products industries because of their low volatile organic contents (VOCs) [13] and good durability [14,15]. Wood often suffers from water damage [16–18] because of the hydrophilicity of its chemical components [19]. Coatings can protect wood from directly contacting water and prevent wood surface from degradation [20–22]. Coating solid wood has the advantage of enhancing its natural beauty and its physical and chemical properties. However, solvent-based coatings can cause damage to the environment and human health [23–26] because of benzene, toluene, xylene, and substances of high carcinogenicity contained in the coatings.

Nanomechanic technology has become the leading option for analyzing micro-mechanical properties of composite interfaces by virtue of its convenient operation and high resolution. Nanomechanic characterization methods are commonly used in scientific research, started by Gindl [27], who was the first to use nanoindentation technology to study the mechanical properties of wood adhesion interface. After publishing their groundbreaking work, the team continued to study the effects of cell cavity filling on the elastic modulus and hardness of cell walls [28–32]. Nanoindentation techniques are used to calculate the mechanical properties of a material after plastic deformation is generated by the pressing action probe. The information recorded at each test point is only over a span of several hundred nanometers or less, which is not conducive to the study of the penetration of adhesives in various layers of the wood cell wall. In analysis of the interactions between loblolly pine and an applied phenolic ("PF") resin, Wang and Marcott [33,34] combined atomic force microscopy and infrared nanospectroscopy to elucidate the permeability of PF resin into wood cell wall forming an interpenetrating polymer network (IPN) structure. Although these techniques obtain mechanical information over a narrow region, this region does not necessarily reflect the mechanical properties of all surfaces. Due to this narrow scope, it remains impossible to quantify interface bond strength across an entire coated bio-based material.

Based on the shortfalls of literature to date, the aim of this work was to use nanoindentation to evaluate the properties of waterborne polycrylic coatings on the surfaces of extractive-containing and extractive-free wood. Waterborne coatings are used for environmental protection of wood, but the drawback to their application is their inefficiency when the wood which it is applied to contains high levels of natural extractives. Further elucidation of the interface between waterborne coating and

wood will be illustrative of the mechanisms that lead to macroscopic defects produced by extractives within coated wood products.

## 2. Experimental

### 2.1. Materials

The Southern pine wood (*Pinus spp.*) was very popular for making furniture for children. The wood used in this study was a 34-year-old southern pine (*Pinus spp.*) from Crossett, AR, USA. The wood blocks with dimensions of 30 mm long × 16 mm wide × 14 mm thick were prepared from the 25th ring of the discs. The moisture content of the wood sample was 6% and the density was 0.5 g/cm$^3$. All nanoindentation wood samples were sapwood and cut from the same growth ring of latewood to avoid artifacts due to natural heterogeneity. The coating material supplied by the Upper Saddle River, NJ, USA was a commercial Minwax waterborne polycrylic protective type, containing alkyl propanols, ethylene glycol, glycol ethers, and 1-methyl-2-pyrrolidinone. The coating had a dissolved solids content of 30%, pH value of 6, and a viscosity of 570 MPa·s at 25 °C.

### 2.2. Sample Preparation by Ethanol-Toluene Extraction

Pine wood samples were cut parallel to the direction of the grain to a length of 12 mm, a width of 8 mm, and a thickness of 7 mm. After cutting, the samples were placed in a soxhlet extraction apparatus and extracted with an ethanol–toluene mixture (1:2 *v:v*) fluxing over 4 h. The spent extraction solvent was discarded and then the material was extracted with ethanol for at least 4 h, until the ethanol spilling over the sample tube became colorless. Next, the wood was removed and air-dried in a fume hood. Ethanol-free wood samples were then transferred to a 7.5 L Florence flask and extracted successively with three 1 L portions of distilled water, heating the flask with each change of water for 1 h in a hot-water bath at 100 °C. After the third extraction with water, the material was filtered on a Buchner funnel and washed with 500 mL of partially boiling distilled water, resulting in production of truly extractive-free wood samples. Finally, the control wood (un-extracted) and extractive-free wood samples were conditioned in a climate (25 °C and 35% relative humidity) for 3 weeks to equilibrate moisture contents up to about 6%.

### 2.3. Coating Process

All the samples were taken from latewood of the same growth ring to avoid artefacts resulting from natural heterogeneity. The surfaces of the prepared wood blocks in both the radial and tangential directions were first smoothed using a microtome before applying the acrylic coating. Three coats of an acrylic-based clear waterborne coating were then sprayed onto the surfaces, with light sanding by 240 mesh sandpaper taking place between coats. The first layer on the wood samples along the grain direction using a brush, air drying samples for two hours, and sanding coated surfaces with 220 grit sandpaper, and then brushing the second coating layer, followed by repeating the air-drying and sanding, then applying the third coating layer. Finally, the coated wood samples were kept on a dust-free plane and then stored in a controlled climate (25 °C and 30% ± 2% relative humidity) for 7 days. The thickness of the coating layer was calculated by the coated sample's thickness subtracted by the original wood's thickness. i.e., the thickness of coating layer was 80 μm. To prepare samples for nanoindentation, the coated wood samples and uncoated wood were cut into a block with a length of 5 mm, a width of 2 mm and a thickness of 1 mm. The cross-section of the coated wood sample was next polished using an ultramicrotome (Leica Ultracut R, LKB-2188, Bromma, Sweden) equipped with a diamond knife. A smooth and well-polished surface with a roughness of less than 100 nm was obtained to ensure adequate contact. The samples were ready for nanoindentation after being affixed to an aluminum mounting [35,36]. Following mounting, the coated wood samples were placed inside the nanoindentation enclosure at least minimum of 24 h prior to experiments to allow equilibration

with the conditions inside the enclosure (25 °C and 30% ± 2% relative humidity). A schematic diagram of sample preparation is shown in Figure 1.

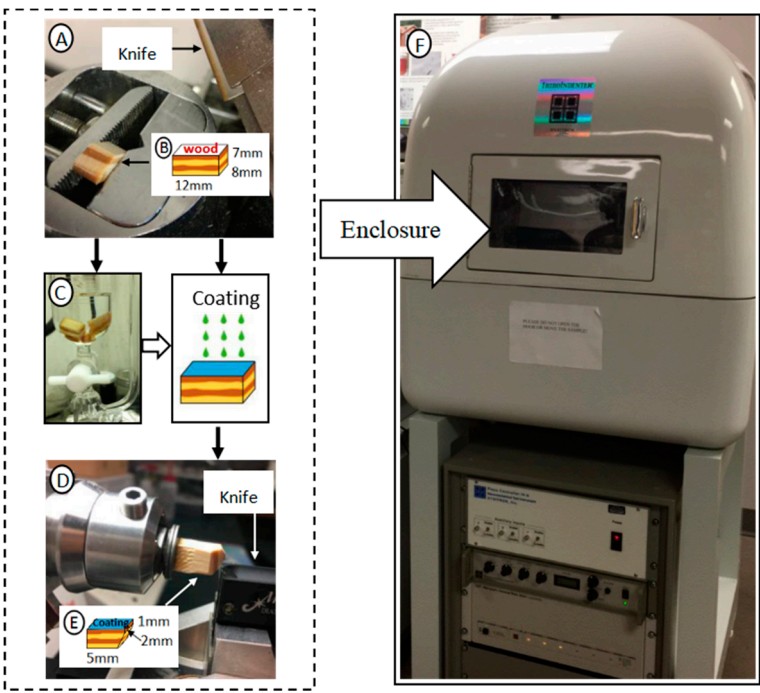

**Figure 1.** Diagram of sample preparation: microtome (**A**); wood sample (**B**); soxhlet extraction apparatus (**C**); ultramicrotome (**D**); wood sample (**E**); and Hysitron TriboIndenter system (**F**).

## 2.4. Nanoindentation

For the nanoindentation test, a Hysitron TriboIndenter system (Hysitron Inc., Minneapolis, MN, USA) equipped with scanning probe microscope (SPM) and a three-sided pyramid diamond indenter tip (Berkovich type) was used. As shown in Figure 2, at least thirty valid indents were performed on the coating layer (C1–C8), the control wood cell wall (W), the wood cell lumen penetrated with coating (WC and PC) at the wood-coating bondline, and on the compound corner-middle lamellae (CCML). Two replicates were examined through selecting 3 or 4 areas (35 µm by 35 µm) within each replicate.

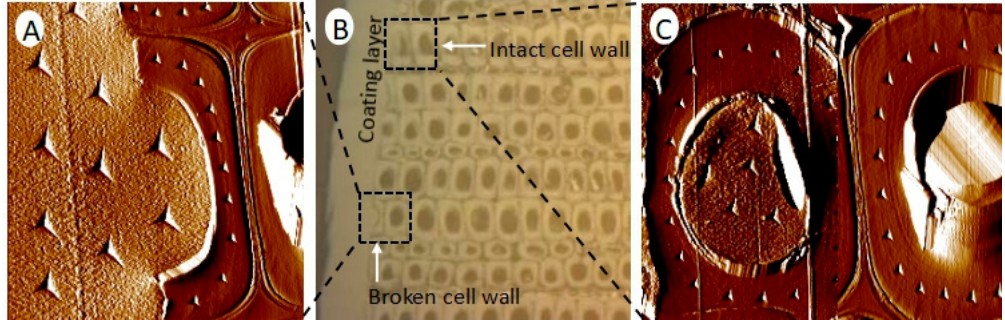

**Figure 2.** Microscopic and SPM images for the coated control wood sample and locations of indents: (**A**,**C**) SPM images of test locations and the indents after characterization by nanoindentation; (**B**) incident light micrograph of the coated layer and the wood cell wall.

Nanoindentation experiments were examined in load-controlled mode using a three-segment load ramp: load application 5 s, holding time 5 s, and unloading time 5 s. The peak load was 400 µN for all indents in the experiments. Figure 3 shows the typical load-displacement curves of samples at

room temperature. The reduced elastic modulus ($E_r$) and hardness ($H$) were calculated according to the method of Oliver and Pharr [37], as follows:

$$E_r = \frac{\sqrt{\pi}}{2\beta} \frac{S}{\sqrt{A}} \tag{1}$$

$$H = \frac{P_{max}}{A_{hc}} \tag{2}$$

where $S$ is initial unloading stiffness, and $\beta$ is a correction factor correlated to the indenter's geometry ($\beta$ = 1.034 for a Berkovich indenter). $P_{max}$ is the peak load, and $A_{hc}$ is the projected contact area at peak load. The data was analyzed by the statistical analysis software SPSS 16.0 and one-way analysis of variance (one-way ANOVA) was used to reach reliable conclusions. Data was also analyzed by post hoc test, which is the test method applied after ANOVA analysis to distinguish significantly different mean values with the Duncan method at 0.05 confidence level [38]. The functional groups of control and extractive-free wood was evaluated by FT-IR (Figure 4), which was important to help us explain the difference of nano-mechanical properties of coatied wood samples. As shown in Figure 5C, the indents were performed on the coating layer with different distance away from wood cell wall coded as C1, C2, C3, C4, C5, C6, C7, and C8, respectively. C1–C6 locations meant the indents examined at coating layer and C7, C8 locations meant coating penetrated and cured in broken cell wall with partial cell structure (PC) and intact cell wall with whole size cell structure (WC), respectively.

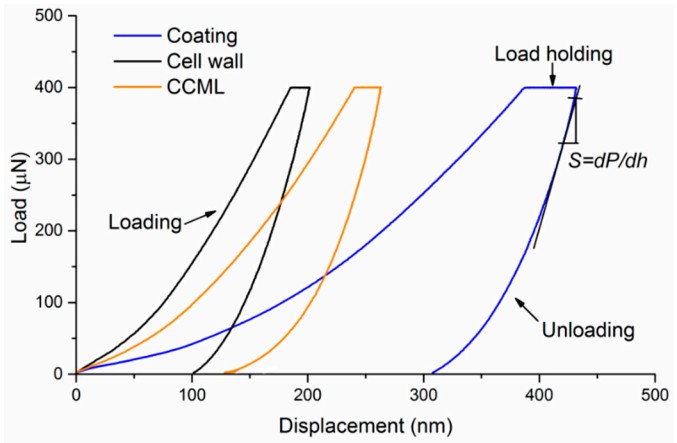

**Figure 3.** Load-displacement curves of acrylic-coated wood samples at room temperature.

### 2.5. FT-IR Analysis

After drying the treated and the untreated wood, a waterborne acrylic coating is applied to the wood surface. The chemical composition and functional groups of the coating surface, the unextracted wood surface, and the extracted wood surface were analyzed using a Bruker FT-IR spectrometer (Avance 300, Brurker Co., Billerica, MA, USA). The emission method was selected with a spectrum scanning frequency of 200 times across the acquisition range of 4000–500 cm$^{-1}$.

### 2.6. Specific Surface Area Analysis

Specific surface area and pore size distribution in samples were tested on an ASAP 2020 specific surface area and aperture analysis tester in the United States. Based on the measured adsorption isotherm of nitrogen under 77 K, the specific surface area was calculated by the BET method and the distribution of medium pore diameter was calculated by the BJH method. The volume of liquid nitrogen was converted to the volume of liquid nitrogen when the relative pressure was 0.99, and the volume of medium pore was obtained by the *t*-plot method.

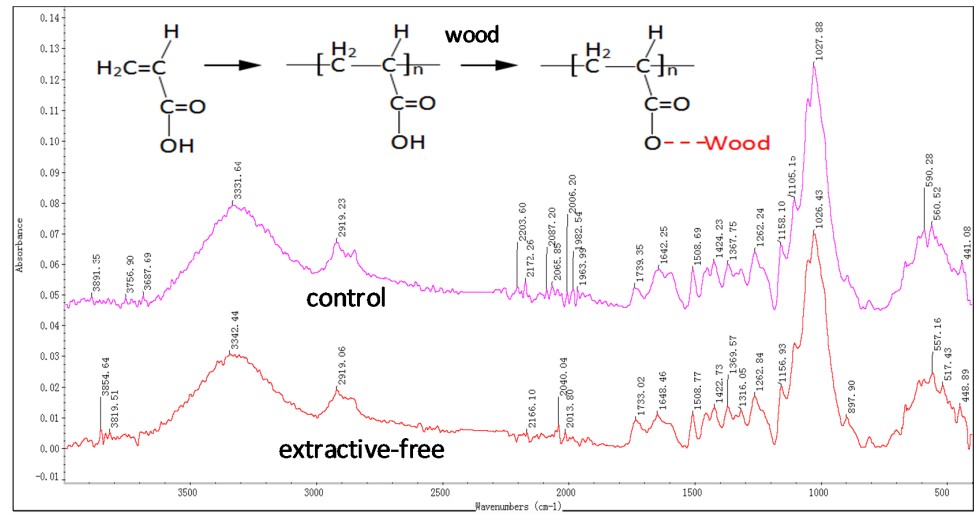

**Figure 4.** Spectra of control and extractive-free wood.

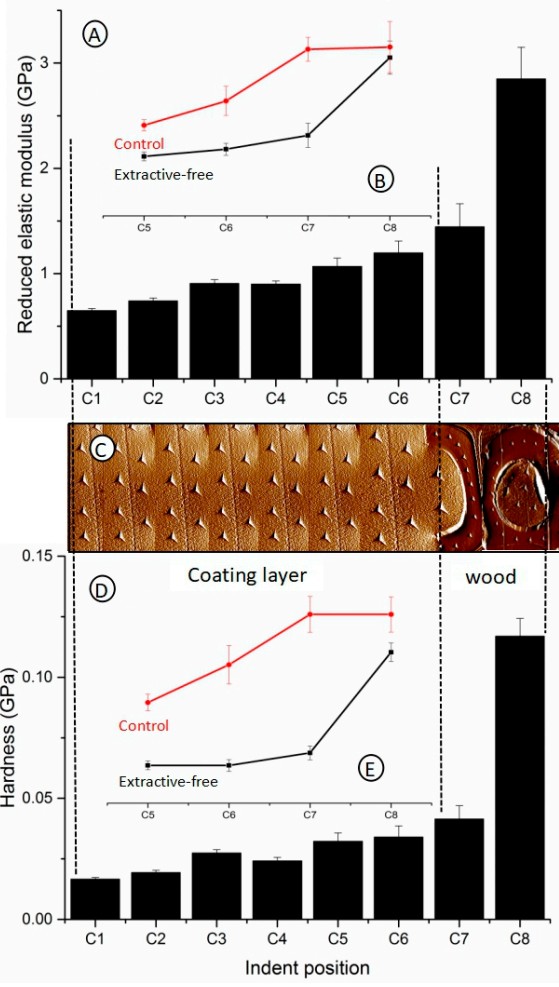

**Figure 5.** Modulus and hardness of water-based acrylic coating layer evaluated by nanoindentation (C1–C6: indents at coating layer; C7 and C8: indents at coating filled in the cell lumen; the inserted figures compare control wood (red line) and extractive-free wood (black line) at C5–C8 locations): (**A**) Reduced elastic modulus of coating layer indented at C1–C8; (**B**) Reduced elastic modulus indented at C5–C8; (**C**) SPM images of coating layer and wood cell wall after nanoindenation test; (**D**) Hardness of coating layer indented atC1–C8; (**E**) Hardness indented at C5–C8.

## 3. Results and Discussion

### 3.1. Rheological Analysis Combined with FT-IR

The FT-IR spectra in Figure 4 shows the differences in functional groups between control and extractive-free wood. For the control wood, multiple absorption peaks appeared between 2000 and 2200 cm$^{-1}$. The presence of these peaks indicates that the wood surface is likely involved with the hydrogen ions in water, showing signs of hydrogen bonding. There was also an absorption peak at 500–600 cm$^{-1}$ attributable to forming the C–1 bond, indicating that the extract in the control wood reacts with the coating which the chemical reaction of phenol-formaldehyde resin molecules with the cell wall. The C=C bond in the waterborne acrylic coating reacts with the OH bond of free water in the surface layers of the wood and opening of this C=C bond leads to the degradation of the performance of the waterborne coating. Differently for the extractive-free wood, there was only one absorption peak between 2000 and 2200 cm$^{-1}$. The presence of a single peak in this region suggests that the extractive-free wood has a certain hydrophobicity that limits the extent of water interaction. In addition, the absorption peak between 500 and 600 cm$^{-1}$ decreased, indicating that the reaction between wood and the coating was weakened. From this we conclude that the C–C bond originally located at 1642 cm$^{-1}$ in the control wood contracted significantly and became C=C bonds in the extractive-free wood. The expansion vibration absorption peak moved to the low wave number at 1648 cm$^{-1}$ and the expansion vibration absorption peak of the O–H band at 500–600 cm$^{-1}$ changed significantly. This indicates that hydrogen bonds are formed between molecules the control wood and waterborne acrylic coatings. Our findings are in agreement with the work performed by Wang et al. [9], who concluded that the chemical reaction between coating and wood enhances the adhesion of coating on wood surface.

### 3.2. Reduced Elastic Modulus and Hardness of Coating Layer

Figure 5A,D compare the elastic modulus ($E_r$)and hardness ($H$) of the coating layer on the extractive-free and control wood surfaces. It can be observed that both of these properties are lowered for the extractive-free samples. In addition, the inserted curves (Figure 5B,E) compare these values for the two samples analyzed. Finally, Figure 5C displays the SPM images of indents on the coating layer and the sub-surface wood cell walls. The $E_r$ and $H$ of the coating layer increase along thickness direction at the confidence level of 0.05 ($p < 0.05$). The average values of $E_r$ and $H$ for coating layer at C1–C6 were 0.9 and 0.03 GPa, respectively, which increased dramatically when moving closer to the wood cell wall. Increases in the reduced elastic modulus at the C7 and C8 positions were observed, resulting in a 59% increase to 1.45 GPa and a 62% to 2.85 GPa, respectively. Moving further inward, significant increases to surface hardness were observed due to the cell wall supporting effect. The $E_r$ and $H$ of the coating layer on the extractive-free wood surface were both lower than those on the control wood surface, which indicated that the extractives in the wood had a positive impact on the mechanics of coating layer. Usually the organic extractives in pine wood, composed of fats and waxes, are able to retard the chemical reaction of phenol-formaldehyde resin molecules with the cell wall [39]. The PF resin in the cell lumen contacted the wood extractives adequately when it flowed through the porous structure of the wood, meaning the chemical nature of the resin in the cell lumen was modified. Furthermore, wood extractives may negatively affect PF resin's ability to infiltrate into the wood. Results showed that the permeability of wood was improved by extraction with cold water, hot water, 1% NaOH, and benzene–ethanol, with the greatest improvement being attributable to alkali and benzene–ethanol extraction. Not only does extraction obviously improve the permeability of wood, but it also results in a more uniform permeability profile. On the other hand, the extractive-free wood has higher wettability due to removal of hydrophobic extractives, which makes it easier for water-based coatings to swell up the wood cell wall and lower $E_r$ and $H$ of compared to control wood. Regarding viscosity of the coating applied, the coating in this work was 568 MPa·s at 25 °C, which made it difficult for the liquid to penetrate into the cell wall. The coating

used by Wang et al. [9] was 200 MPa·s at 25 °C, which may explain the greater coating success of that work. Increased surface areas value in extractive-free wood measured by BET also verified our inference, the BET specific surface area of controlled wood is 0.701 $m^{-2}$·g, while the value of extracted wood is 22.3 $m^{-2}$·g. The increase of the specific surface area of extractive-free wood leads to increased porosity, which increases the contact area between coating and wood. This enhances the permeation of water and dissolved coating molecules into wood cell walls, resulting in more efficient coating.

### 3.3. Reduced Elastic Modulus and Hardness of Wood Cell Wall

The $E_r$ and $H$ of control and extractive-free wood cell wall were investigated. As presented in Figure 6, the $E_r$ and $H$ values varied over different locations, e.g., the control cell wall, the coating penetrated intact cell wall (WC) and broken cell wall (PC), and compound corner middle lamella (CCML). For extractive-free wood, the $E_r$ in W, WC, PC and CCML was 17.57, 15.64, 12.62 and 5.99 GPa, moreover, the hardness was 0.43, 0.39, 0.42 and 0.35 GPa, respectively. The $E_r$ in W, WC, PC, and CCML was 17.15, 15.13, 12.53 and 6.36 GPa for control wood, respectively. The hardness was 0.42, 0.39, 0.41 and 0.40 GPa, respectively. When the microtome and ultramicrotome were used to smooth wood surface for better coating quality, the cell wall was broken (as shown in Figure 2A,B). The coating swill flows into the open cell and cell lumen from the pith. The cell wall will be swelled up by the water in the coating, which resulted in the decrease to its mechanical properties [40]. This result is opposite of the findings of our previous work, which showed that the cell wall in a resin glueline region had significantly improved mechanical properties compared to the cell wall matter further away from glueline [9]. The coating usually has larger molecule weight than resin, which can retard its penetration into the wood cell lumen and cell wall and decrease the cost of manufacture. The $E_r$ and $H$ of extractive-free wood were similar to the ones of control wood cell wall. The $E_r$ of wood cell walls were all significantly higher than that of CCML ($p < 0.05$) in both extractive-free and control wood cell wall, and this observation is in agreement with the literature data [41–43]. However, there was no distinct differences in hardness between the cell wall and CCML. When comparing $E_r$ and $H$ for control and extractive-free, which were 15% and 6% lower in extractive-free sample than those in control wood cell walls, respectively. This phenomenon might be attributed to the loss of extractives, as the CCML hardness decreased obviously in agreement with our reasoning.

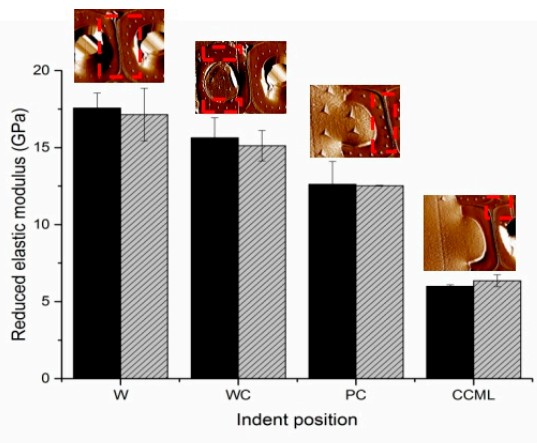

**Figure 6.** *Cont.*

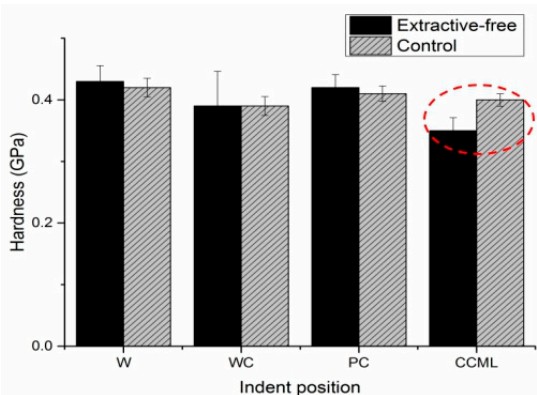

**Figure 6.** Reduced elastic modulus and hardness of wood cell wall and compound corner lamella (the inserted scanning probe microscope (SPM) images showed the indents at wood cell wall (W, WC and PC) and compound corner middle lamella (CCML)).

## 4. Conclusions

- The elastic modulus and the hardness of acrylic coating decreased by 16% and 5% for extractive-free wood compared with the control wood, indicating that the extracted material prevented the coating from entering the wood cell wall cavities. However, the wood hardness in CCML layer decreased significantly, indicating that the water-based coating entered into the wood cell when the wood surface was extracted, resulting in a decline to its mechanical properties.
- The specific surface area of the extractive-free wood increased, and the specific surface area was positively correlated with the adsorption capacity, indicating that compared with the control wood, the adsorption ability of the extractive-free wood to the waterborne coating was enhanced.
- The surface characteristics of extractive-free wood changed greatly compared to a control. FT-IR analysis showed that the O–H bond in waterborne coatings combined with the wood extract to form stable structures, indicating that the extractive-free wood reacts with the waterborne coating and enhance the performance of the coating on the wood.

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

**Conflicts of Interest:** The authors declare no conflicts of interest.

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
