# Peer review of "The Implication of Benzene–Ethanol Extractive on Mechanical Properties of Waterborne Coating and Wood Cell Wall by Nanoindentation"

_coatings, doi:10.3390/coatings9070449_

Reviewer 1 Report

This paper focuses upon the mechanical properties of waterborne polyacrylic coating on control and extractive-free wood surfaces investigated by nanoindentation. The topic is very interesting from scientific point of view. In last year’s scientists used nanoindentation as a tool for wood and coatings analysis. The topic is of current interest and the testing reported could produce valuable outcomes, anyway the research presents the following issues:

- To the keywords Authors should add wood species – Pine wood

- There is a lack of papers concerning with the lacquer products and influence of the properties of coatings formed on the pine wood surface.

- The selection of wood species, acrylic lacquer product should be explained. Why Authors used pine wood (Pinus spp.) for the experiments? Basic properties of pine wood used for the experiments are necessary for further analysis! For example, density and kind of wooden parts – sapwood or heartwood, annual growth rates should be given.

- There is a lack of information about the finishing parameters – only the coatings thickness. What about the conditioning time between lacquer application and investigations?

- How many samples Authors used for experiments? How many repetitions have been made?

- There is a lack of the statistical estimation.

- Why Authors used the relative humidity 30±2 % (always for coatings RH 65±5% or 50±5% is used)?

- In the conclusions Authors wrote: “…This resulted in the waterborne coating having an easier route to penetrate into the wood cell wall and cell cavities …”. Authors did not investigate penetration of the acrylic lacquer product into wood.

Paper can be published after minor changes and additions.

Author Response

I am pleased to resubmit for the revised version of manuscript entitled “The implication of benzene-ethanol extractive on mechanical properties of waterborne coating and wood cell wall by nanoindentation”. Thank you for reading our manuscript and reviewing it. Those comments are all valuable and very helpful for revising and improving our paper. We have revised our manuscript carefully and have made correction which we hope meet with approval. So we have sent the revised manuscript and have highlighted changes by using the word track modeThe main corrections in the paper and the responds to the reviewers’ comments are as following:

Responds to the reviewers’ comments:

Response to reviewer 1:

Comments and Suggestions for Authors

This paper focuses upon the mechanical properties of waterborne polyacrylic coating on control and extractive-free wood surfaces investigated by nanoindentation. The topic is very interesting from scientific point of view. In last year’s scientists used nanoindentation as a tool for wood and coatings analysis. The topic is of current interest and the testing reported could produce valuable outcomes, anyway the research presents the following issues:

- To the keywords Authors should add wood species – Pine wood

Answer: The keywords “Pine wood” was add in the keyword list.

- There is a lack of papers concerning with the lacquer products and influence of the properties of coatings formed on the pine wood surface.

Answer: The references were added as “Waterborne coatings like UV-cured, acrylic varnish, etc. are increasingly used in wood products industries because of their low volatile organic contents (VOCs) [13] and good durability [14-15]. Wood often suffers from water damage [16-18] because of the hydrophilicity of its chemical components [19]. Coatings can protect wood from directly contacting with water and prevent wood surface from degradation [20-22]. Coating solid wood has the advantage of enhancing its natural beauty and its physical and chemical properties. But solvent-based coatings can cause damage to the environment and human health [23-26] because of benzene, toluene, xylene, and substances of high carcinogenicity contained in the coatings.”

13. Vardanyan, V.; Galstian, T.; Riedl, B. Characterization of cellulose nanocrystals dispersion in varnishes by backscattering of laser light, J. Coat. Technol. Res. 2015, 12: 647-656.

14. Custódio, J.E.P.; Eusébio, M.I. Waterborne acrylic varnishes durability on wood surfaces for exterior exposure. Prog. Org. Coat. 2006, 56: 59-67.

15. Miklečić, J.; Blagojević, S.L.; Petrič, M.; Jirouš-Rajković, V. Influence of TiO2 and ZnO nanoparticles on properties of waterborne polyacrylate coating exposed to outdoor conditions. Prog. Org. Coat. 2015, 89: 67-74.

16. Huang, C.; Chu, Q.; Xie, Y.; Li, X.; Jin, Y.; Min, D.; Yong, Q. Effect of kraft pulping pretreatment on the chemical composition, enzymatic digestibility, and sugar release of moso bamboo residues, Bioresources 2015, 10: 240-255.

17. Huang, C.; He, J.; Du, L.; Min, D.; Yong, Q. Structural characterization of the lignins from the green and yellow bamboo of bamboo culm (Phyllostachys pubescens), J. Wood Chem. Technol. 2016, 36: 157-172.

18. Mahltig, B.; Swaboda, C.; Roessler, A.; Boettcher, H. Functionalising wood by nanosol application. J. Mater. Chem. 2008, 18: 3180-3192.

19. Xiong, X.; Bao, Y.; Liu, H.; Zhu, Q.; Lu, R.; Miyakoshi, T. Study on mechanical and electrical properties of cellulose nanofibrils/graphene-modified natural rubber. Mater. Chem. Phys. 2019, 223 (2): 535-541.

20. Huang, C.; Su, Y.; Shi, J.; Yuan, C.; Zhai, S.; Yong, Q. Revealing the effects of centuries ageing on the chemically structural features of lignin in archaeological fir woods. New J. Chem. 2019, 43: 3520-3528.

21. Grüneberger, F.; Künniger, T.; Zimmermann, T.; Arnold, M. Rheology of nanofibrillated cellulose/acrylate systems for coating applications. Cellulose 2014, 21: 1313-1326.

22. Sonderegger, W.; Glaunsinger, M.; Mannes, D.; Volkmer, T.; Niemz, P. Investigations into the influence of two different wood coatings on water diffusion determined by means of neutron imaging. Eur. J. Wood Prod. 2015, 73: 1-7.

23. Cataldi, A.; Corcione, C.E.; Frigione, M.; Pegoretti, A. Photocurable resin/nanocellulose composite coatings for wood protection. Prog. Org. Coat. 2017, 106: 128-136.

24. Kaboorani, A.; Auclair, N.; Riedl, B.; Landry, V. Physical and morphological properties of UV-cured cellulose nanocrystal (CNC) based nanocomposite coatings for wood furniture. Prog. Org. Coat. 2016, 93: 17-22.

25. Qi, Y.; Shen, L.; Zhang, J.; Yao, J.; Lu R.; Miyakoshi T. Species and release characteristics of VOCs in furniture coating process, Environ. Pollut. 2019, 245: 810-819.

26. Zhao, Z.; Hayashi, S.; Xu, W.; Wu, Z.; Tanaka, S.; Sun, S.; Zhang, M.; Kanayama, K.; Umemura, K. A novel eco-friendly wood adhesive composed by sucrose and ammonium dihydrogen phosphate. Polymers 2018, 10: 1251-1265.

- The selection of wood species, acrylic lacquer product should be explained. Why Authors used pine wood (Pinus spp.) for the experiments? Basic properties of pine wood used for the experiments are necessary for further analysis! For example, density and kind of wooden parts – sapwood or heartwood, annual growth rates should be given.

Answer: The pine wood (Pinus spp.) was very poplar used for making furniture for Children. The wood used in this study was a 34-year-old southern pine (Pinus spp.) from Crossett, Arkansas, USA. The wood blocks with dimensions of 30 mm long × 16 mm wide × 14 mm thick were prepared from the 25th ring of the discs. The moisture content of the wood sample was 6 % and the density was 0.5 g/cm3. All nanoindentation wood samples were sapwood and cut from the same growth ring of latewood to avoid artifacts due to natural heterogeneity. The coating material supplied by the Upper Saddle River, New Jersey, USA was a commercial Minwax waterborne polycrylic protective type, containing alkyl propanols, ethylene glycol, glycol ethers, and 1-methyl-2-pyrrolidinone.

- There is a lack of information about the finishing parameters – only the coatings thickness. What about the conditioning time between lacquer application and investigations?

Answer: The first layer on the wood samples along the grain direction using a brush, air drying samples for two hours, and sanding coated surfaces with 220 grit sandpaper, and then brushing the second coating layer, followed by repeating the air-drying and sanding, then applying the third coating layer. Finally, the coated wood samples were kept on a dust-free plane and then stored in a controlled climate (25 ℃ and 30 ± 2 % relative humidity) for 7 days.

- How many samples Authors used for experiments? How many repetitions have been made?

Answer: At least thirty valid indents were made on each of two locations, coating layer and wood cell walls, to ensure 30 points for the calculation of mean values of reduced elastic modulus and hardness. Two replicates were examined through selecting 3 or 4 areas (35 µm by 35 µm) within each replicate.

- There is a lack of the statistical estimation.

Answer: The data was analyzed by the statistical analysis software SPSS 16.0 and one-way analysis of variance (one-way ANOVA) was used to reach reliable conclusions. Data was also analyzed by post hoc test, which is the test method applied after ANOVA analysis to distinguish significantly different mean values with the Duncan method at 0.05 confidence level [38].

- Why Authors used the relative humidity 30±2 % (always for coatings RH 65±5% or 50±5% is used)?

Answer: we kept the conditions the same between the examined smaples and nanoindentation conditions.

- In the conclusions Authors wrote: “…This resulted in the waterborne coating having an easier route to penetrate into the wood cell wall and cell cavities …”. Authors did not investigate penetration of the acrylic lacquer product into wood.

Answer: This sentence was delted.

We appreciate for Editor and Reviewers’ warm work earnestly, and hope that the correction will meet with approval. Once again, thank you very much for your comments and suggestions.

Yours sincerely,

Yan Wu and Feng Yang

Reviewer 2 Report

Dear Authors, 

I have read the article entitled „The implication of benzene-ethanol extractive on mechanical properties of waterborne coating and wood cell wall by nanoindentation”.

I have only few observations. Based on them the paper can be briefly revised:

Page 2, line 69: Gindl and Gupta [13] does not appear in the reference list

Page 2 lines 76-79: the text is too long and it is difficult to understand, please split it in two sentences

Page 2 line 76: Wang et al. [19-20] is not correct, ref [19] is right but ref [20] is not, see the reference list

Page 5 line 157: Fig. 4C is mentioned here, please check

Page 6 line 196: the reference Wang et al. [8] does not appear the same in the reference list

I hope my revision was of help.

Author Response

I am pleased to resubmit for the revised version of manuscript entitled “The implication of benzene-ethanol extractive on mechanical properties of waterborne coating and wood cell wall by nanoindentation”. Thank you for reading our manuscript and reviewing it. Those comments are all valuable and very helpful for revising and improving our paper. We have revised our manuscript carefully and have made correction which we hope meet with approval. So we have sent the revised manuscript and have highlighted changes by using the yellow colour. The main corrections in the paper and the responds to the reviewers’ comments are as following:

Responds to the reviewers’ comments:

Reviewer 2 Comments: 

1) Page 2, line 69: Gindl and Gupta [13] does not appear in the reference list

Answer: We had a mistake with the author of reference [13], we already delete it. Now changed into [27] Gindl, W.; Dessipri, E.; Wimmer, R. Using UV-microscopy to study diffusion of melamine-urea-formaldehyde resin in cell walls of spruce wood, Holzforschung 2002, 56(1): 103-107.

2) Page 2 lines 76-79: the text is too long and it is difficult to understand, please split it in two sentences

Answer: The sentence “In analysis of the ineractions between loblolly pine and an applied phenolic (“PF”) resin, Wang and Marcott [33-34] combined atomic force microscopy and mechanical test module spain stakingly elucidated wood cell wall’s permeability to an interpenetrating polymer network (IPN) structure while also obtaining the changes induced to the mechanical properties of the newly coated material.” was rewritten by “In analysis of the ineractions between loblolly pine and an applied phenolic (“PF”) resin, Wang and Marcott [19-20] combined atomic force microscopy and mechanical test module spain. Stakingly elucidated wood cell wall’s permeability to an interpenetrating polymer network (IPN) structure while also obtaining the changes induced to the mechanical properties of the newly coated material.”.

3) Page 2 line 76: Wang et al. [19-20] is not correct, ref [19] is right but ref [20] is not, see the reference list.

Answer: The sentence "Wang et al. [19-20] combined atomic force microscopy and mechanical test module spain." was rewritten by "Wang and Marcott [33-34] combined atomic force microscopy and mechanical test module spain."

33. Wang, X.; Deng, Y.; Li, Y.; Kjoller, K.; Roy, A.; Wang, S. In situ identification of the molecular-scale interactions of phenol-formaldehyde resin and wood cell walls using infrared nanospectroscopy, RSC Adv 2016, 6: 76318-76324.

34. Marcott, C.; Lo, M.; Hu, Q.; Dillon, E.; Prater, C.B. Nanoscale infrared spectroscopy of polymer composites, Am Lab 2014, 46(3): 23-25.

4) Page 5 line 157: Fig. 4C is mentioned here, please check.

Answer: We changed the Fig. 4C into Fig. 5C.

5) Page 6 line 196: the reference Wang et al. [8] does not appear the same in the reference list.

Answer: The sentence "Our findings are in agreement with the work performed by Wang et al. [8]" was rewritten by "Our findings are in agreement with the work performed by Wang et al. [9]"

We appreciate for Editor and Reviewers’ warm work earnestly, and hope that the correction will meet with approval. Once again, thank you very much for your comments and suggestions.

Yours sincerely,

Yan Wu and Feng Yang
